# A NEW METHOD OF REGION EMBEDDING FOR TEXT CLASSIFICATION

**Chao Qiao**[*‡]**, Bo Huang**[†‡]**, Guocheng Niu**[‡]**, Daren Li**[‡]**,**
**Daxiang Dong**[‡§]**, Wei He**[‡]**, Dianhai Yu**[‡§]**, Hua Wu**[‡§]

[‡] Baidu Inc., Beijing, China
[§] National Engineering Laboratory of Deep Learning Technology and Application, China
```
{qiaochao, huangbo02, niuguocheng, lidaren,
daxiangdong, hewei06, yudianhai, wu_hua}@baidu.com
```

## ABSTRACT

To represent a text as a bag of properly identified "phrases" and use the representation for processing the text is proved to be useful. The key question here is how to identify the phrases and represent them. The traditional method of utilizing n-grams can be regarded as an approximation of the approach. Such a method can suffer from data sparsity, however, particularly when the length of n-gram is large. In this paper, we propose a new method of learning and utilizing task-specific distributed representations of n-grams, referred to as "region embeddings". Without loss of generality we address text classification. We specifically propose two models for region embeddings. In our models, the representation of a word has two parts, the embedding of the word itself, and a weighting matrix to interact with the local context, referred to as local context unit. The region embeddings are learned and used in the classification task, as parameters of the neural network classifier. Experimental results show that our proposed method outperforms existing methods in text classification on several benchmark datasets. The results also indicate that our method can indeed capture the salient phrasal expressions in the texts.

## 1 INTRODUCTION

Text classification is an important task for many applications, including topic categorization, search query classification, and sentiment analysis, which has been studied for years. A simple yet effective approach for text classification is to represent documents as bag-of-words, and train a classifier on the basis of the representations using methods such as logistic regression, support vector machines (Joachims, 1998; Fan et al., 2008), and naive Bayes (McCallum et al., 1998). Although bag-of-words methods are effective and efficient, they also have limitations. The representations do not take into account the word order information which has been proved to be useful at least in some applications such as sentiment analysis (Pang et al., 2002).

To make effective use of word order information for text classification, people have traditionally exploited n-grams, i.e., short sequences of words in the texts. Previous work shows that the use of n-grams is effective in the text classification task (Pang et al., 2002; Wang & Manning, 2012; Joulin et al., 2016). Although n-grams are very useful, they have certain limitations. 1) The number of n-grams increases exponentially when the length of n-gram $n$ increases. This makes it difficult to exploit large n-grams (e.g., $n > 4$). 2) Since the number of parameters in an n-gram model is very large, the estimation of the parameters usually suffers from the data sparsity problem.

Recently, the method of FastText has been proposed (Joulin et al., 2016), which can learn and use distributed embeddings of n-grams. More specifically, the embedding of an n-gram is defined as a low-dimensional vector representation of the n-gram. Note that the n-grams in a vocabulary can also be represented as one-hot vectors Wang & Manning (2012).

---

[*]chao.qiao@outlook.com
[†]bohuang0321@gmail.com

In this paper, we propose to learn embeddings of n-grams for a specific task (e.g., classification), which are more compact and thus easy to obtain. We call the embeddings region embeddings, following the work in Johnson & Zhang (2015). Our method significantly differs from their method, however, in the sense that the region embeddings in our method are task-dependent and acquired from supervised learning, while those in their method are task-independent and acquired from unsupervised learning. Our method is also largely different from FastText, as it learns richer models for region embeddings.

Intuitively, the meaning of a word is defined by the meaning of itself as well as the meanings of words in the surrounding context. The *extended* embedding of a word in an n-gram thus consists of two parts, the embedding of the word itself and a matrix to interact with the local context, named "local context unit". The embedding of a word is a column vector, and the local context unit of a word is a matrix in which the columns are used to interact with words in the local context. The region embedding of an n-gram is then constructed by the extended embeddings of all words in the n-gram. In this paper, we introduce two models for region embeddings.

For the text classification task, a document is viewed as a bag of region embeddings, and the bag of region embeddings is fed into a classifier. The parameters of the local context units and word embeddings are trained together with the parameters of the classifier which is a fully connected neural network. Our models achieve better results than the state-of-the-art methods on several benchmark datasets of text classification. Experiments show that our proposed models can really capture important information for the task.

## 2 RELATED WORK

Text classification has been studied for years, traditional approaches focused on feature engineering and using different types of machine learning algorithms. For feature engineering, bag-of-words features are efficient and popular. In addition, the hand-crafted n-grams or phrases are added to make use of word order in text data, which has been shown effective on Wang & Manning (2012). For machine learning algorithms, linear classifiers are widely used, such as naive bayes (McCallum et al., 1998), logistic regression and support vector machines (Joachims, 1998; Fan et al., 2008). However, these models commonly suffer the data sparsity problem.

Recently, several neural models have been proposed, the pre-trained word embeddings of *word2vec* (Mikolov et al., 2013) have been widely used as inputs to deep neural models such as recursive tensor networks (Socher et al., 2013). On the other hand, some simple and efficient models which can directly learn task specific word embeddings or fine-tune on pre-trained word embeddings have been proposed recently, such as Deep Averaging Networks (Iyyer et al., 2015), FastText (Joulin et al., 2016). Several neural models have been proposed to make use of word order information, most models are based on convolutional neural network (CNN) (Kim, 2014; Johnson & Zhang, 2014; Zhang et al., 2015) and recurrent neural network (RNN) (Tang et al., 2015; Lai et al., 2015; Yogatama et al., 2017). More recently, the Transformer (Vaswani et al., 2017), a sequence transduction model based solely on attention mechanisms has been proposed. Although Transformer was not designed for the text classification task, it has similarities with our work. In the rest of this section, we will briefly introduce FastText, CNN and Transformer, which are the most relevant to our work.

**FastText**  FastText averages the word embeddings to represent a document, and uses a full connected linear layer as the classifier. The word embeddings are trained for each task specifically. To utilize the local word order information of small regions, FastText uses hand-crafted n-grams as features in addition to single words. With the simple architecture, FastText has been proved to be effective and highly efficient on text classification tasks. Similarly, our models use bag of region embeddings to represent a document, and use the same linear classifier. Differently, our models directly learn the semantics of regions based on word sequence, hand-crafted features are not required.

**CNN**  CNN is a feed-forward network with convolutional layers interleaved with pooling layers, which are originally used for image processing tasks. For natural language processing, words are commonly converted to vectors. CNN directly applies convolutional layer on word vectors, both word vectors and the shared (word independent) kernels are the parameters of CNN, which can be

learned to capture the predictive structures of small regions. The essence of CNN is to learn embeddings for small fixed size regions, each kernel of the convolutional layer tries to capture a specific semantic or structural feature. Our purpose is similar with CNN, which tries to learn task specific representations of regions. Unlike CNN, we apply local context units on word vectors, which are word dependent, moreover, the convolution kernels extract the predictive features by applying convolution operation on word sequences, while we use local context units as distinct linear projection functions on context words in corresponding relative positions to get region representations.

**Transformer** Vaswani et al. (2017) proposed a sequence transduction model, the Transformer, based solely on attention mechanisms. Both Transformer and our method can capture word order information without any CNN or RNN component, and the scalar form of context units (introduced in our ablation experiments) can be regarded as a kind of local attention. There are also some differences here: the motivation we proposed local context units is to address word specific influence between word and its context, while Vaswani et al. (2017) has proposed a parallelable sequence transduction framework based entirely on attention; To utilize position information, in our method, words are interacted with context words at different relative positions by corresponding columns in their context units, while Transformer use fixed $\sin$ and $\cos$ function based position encoding.

## 3 METHOD

In this paper, we focus on learning the representations of small text regions which preserve the local internal structural information for text classification. The regions in a document can be considered as fixed length contiguous subsequences of the document. More specifically, with $w_i$ standing for the $i$-th(starting from 0) word of the document, we use $region(i, c)$ to denote the $2 \times c + 1$ length region with middle word $w_i$. For instance, given a sentence such as *The food is not very good in this hotel*, $region(3, 2)$ means the subsequence *food is not very good*.

In this work, we use the interactions between words and their local context based on word embeddings as well as the local context units to produce region embeddings. In the rest of this section, we will introduce the local context units firstly, and two architectures to generate the region embeddings through local context units will be introduced, finally we will introduce how we use the region embeddings on text classification.

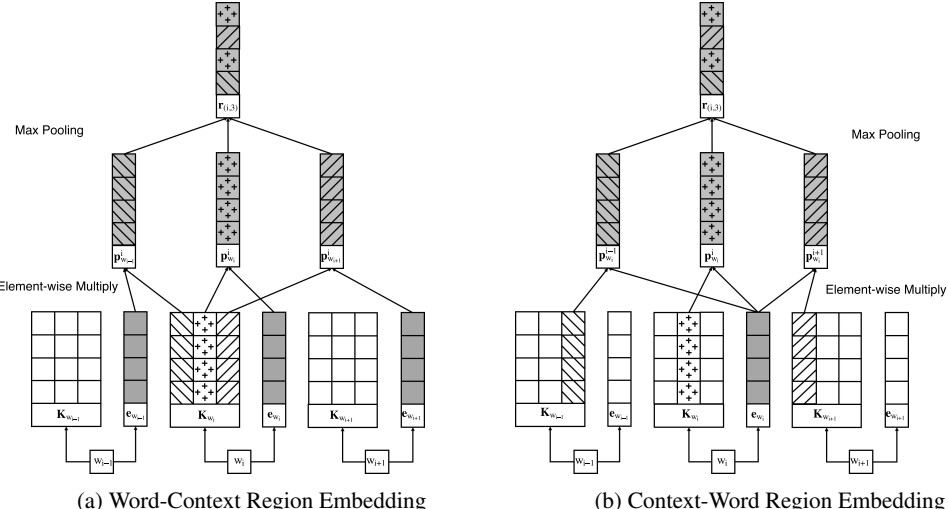

(a) Word-Context Region Embedding    (b) Context-Word Region Embedding

Figure 1: Architectures of region embedding using local context units in different perspectives

### 3.1 LOCAL CONTEXT UNIT

In natural language processing, words are commonly converted to low dimensional vectors(word embeddings) as the inputs to neural networks. More formally, the embedding $\mathbf{e}_w$ of word $w$ is

represented by a column in a matrix $\mathbf{E} \in \mathcal{R}^{h \times v}$ with a look up layer, where $v$ is the size of the vocabulary, $h$ is the embedding size.

To utilize the information of words' relative positions and local context, we learn a local context unit for each word in addition to the word embedding, and both the unit and word embedding are learned as model parameters. Formally, we define the local context unit $\mathbf{K}_{w_i} \in \mathcal{R}^{h \times (2 \times c+1)}$ of $w_i$ as a matrix which can be looked up in the tensor $\mathbf{U} \in \mathcal{R}^{h \times (2 \times c+1) \times v}$ by $w_i$'s index in the vocabulary.

Each column in $\mathbf{K}_{w_i}$ can be used to interact with the context word in corresponding relative position of $w_i$. In fact, the columns of a unit matrix can be regarded as distinctive linear projection functions on the embeddings of words in the local context. The parameters of these projection functions(i.e., columns of each unit matrix) can be learned to capture the semantic and syntactic influence of the word to its context. Word embeddings are used as inputs to the projection functions, and we call the outputs projected word embeddings.

Formally, let $\mathbf{p}^i_{w_{i+t}}$ be the projected word embedding of $w_{i+t}$ in $i$-th word's view, and $\mathbf{K}_{w_i,t}$ be the $(c+t)$-th column in $\mathbf{K}_{w_i}$ $(-c <= t <= c)$, given the unit $\mathbf{K}_{w_i}$ of $w_i$ and the embedding $\mathbf{e}_{w_{i+t}}$ of $w_{i+t}$, we use an element-wise multiplication(denoted by $\odot$) to compute $\mathbf{p}^i_{w_{i+t}}$:

$$\mathbf{p}^i_{w_{i+t}} = \mathbf{K}_{w_i,t} \odot \mathbf{e}_{w_{i+t}} \tag{1}$$

For a context word in a particular relative position of $w_i$, there is a corresponding linear projection function(a particular column of $\mathbf{K}_{w_i}$), thus our proposed local context units can utilize the local ordered word information in a novel way. Note that the middle column $\mathbf{K}_{w_i,0}$ of $\mathbf{K}_{w_i}$ can be regarded as a linear projection function on $\mathbf{e}_{w_i}$ itself, which transforms $\mathbf{e}_{w_i}$ to the same space as other projected embeddings.

## 3.2 Word-context region embedding

We proposed two architectures to perform the region embedding from different perspectives. We consider the semantics of a given region is derived from the mutual influences of the words in this region. In this paper, the regions can be regarded as snapshots of a window sliding on a document, whose middle words are contiguous, hence we can compose the semantics of a give region only by the middle word's influences on the context words, or the context words' influences on the middle word.

In the first proposed architecture, we focus on addressing the middle word's influences on the context words. For example, in the sentence *The food is not very good in this hotel*, the occurrence of word *not* might bring a semantic reversal to the local region.

We use the local context unit of the middle word and the original word embeddings in a region to perform the projected embeddings in a word-to-context view, where the projected embeddings can reflect the middle word's influences on the context words. Once the projected embeddings are obtained, a max pooling operation is applied to extract the most predictive features in the region. The output of the max pooling operation can be regarded as a task related region embedding in a word-to-context view, i.e. Word-Context region embedding.

Formally, we use the context unit $\mathbf{K}_{w_i}$ of middle word $w_i$ and embeddings of all words in a region $region(i, c)$ to compute the projected embedding matrix by equation (1), then the Word-Context region embedding $\mathbf{r}_{(i,c)}$ can be obtained through a max pooling operation on the projected embedding matrix:

$$\mathbf{r}_{(i,c)} = max([\mathbf{p}^i_{w_{i-c}} \quad \mathbf{p}^i_{w_{i-c+1}} \quad \cdots \quad \mathbf{p}^i_{w_{i+c-1}} \quad \mathbf{p}^i_{w_{i+c}}]) \tag{2}$$

where $max$ standing for the max pooling operation on the column dimension of the input matrix. Finally, we get $\mathbf{r}_{(i,c)}$ as a vector representation of $region(i, c)$ with dimension $h$. Figure 1a shows the details of the first model architecture.

For instance, in the sentence *The food is not very good in this hotel*, the projected word embeddings in the $region(3, 2)$ are composed by the element-wise multiplications between columns in the local

context unit of *not* and word embeddings of *food*, *is*, *not*, *very* and *good*. The embedding $\mathbf{r}_{3,2}$ of $region(3, 2)$ can be obtained by max pooling on the projected word embedding matrix.

### 3.3 CONTEXT-WORD REGION EMBEDDING

The second architecture goes as a different view, which addresses the local context words' influences on the middle word in the region, and we call this Context-Word region embedding. Similarly, for a $region(i, c)$, the projected embeddings are computed by the original word embedding of the middle word and the context units of all words in the region, then the Context-Word region embedding can be obtained by a max pooling operation through the column dimension of the projected embedding matrix:

$$\mathbf{r}_{(i,c)} = max([\mathbf{p}_{w_i}^{i-c} \quad \mathbf{p}_{w_i}^{i-c+1} \quad ... \quad \mathbf{p}_{w_i}^{i+c-1} \quad \mathbf{p}_{w_i}^{i+c}]) \tag{3}$$

Figure 1b shows the details of the second model architecture. our two models take different ways to produce the projected word embeddings, the Word-Context model uses context units of middle words and word embeddings of context words, while the Context-Word model uses context units of context words and word embeddings of the middle word.

### 3.4 REGION EMBEDDING FOR TEXT CLASSIFICATION

For text classification, documents are usually variable-sized, which need to be represented as fixed size vectors. In order to show the effectiveness of our proposed region embedding models, we just sum up the embeddings of all regions to represent a document, and feed it to an upper Full-Connected layer for text classification task.

Formally, the model can be represented as following:

$$f(\mathbf{x}; \mathbf{E}, \mathbf{U}, \mathbf{W}, \mathbf{b}) = g(\mathbf{W}\sigma(\sum_{i=0}^{n} \mathbf{r}_{(i,c)}) + \mathbf{b}) \tag{4}$$

where $\mathbf{x}$ denotes the input text sequence, $\mathbf{W}$ and $\mathbf{b}$ denote the weight matrix and bias of the fully connected layer respectively, g denotes the softmax function of the output layer, $\sigma$ denotes the soft-sign function and $n$ denotes the number of regions in a document, $\mathbf{r}$ is the region embedding which can be computed by the equation (2) or (3). $\mathbf{E}$, $\mathbf{U}$, $\mathbf{W}$ and $\mathbf{b}$ can be updated in the training period.

## 4 EXPERIMENTS

We report experiments with proposed models in comparison with previous models.

### 4.1 DATASETS

We use publicly available datasets from Zhang et al. (2015) to evaluate our models. There are in total 8 text classification datasets, corresponding to sentiment analysis, news classification, question-answer, ontology extraction tasks, respectively. Table 1 shows the descriptive statistics of datasets used in our experiments. To guarantee comparable indications, same evaluation protocol of Zhang et al. (2015) is employed.

### 4.2 BASELINES

Our models are compared with several widely used supervised text classification models. We report the n-grams and TFIDF baselines from Zhang et al. (2015), as well as the character level convolutional model (char-CNN) of Zhang & LeCun (2015), the character based convolution recurrent network (char-CRNN) of Xiao & Cho (2016), the very deep convolutional network (VDCNN) of Conneau et al. (2016), the Discriminative LSTM (D-LSTM) of Yogatama et al. (2017) and the bigram FastText (bigram-FastText) of Joulin et al. (2016).

Table 1: Statistics of Datasets

| Dataset | Classes | Average Lengths | Train Samples | Test Samples | Tasks |
|---|---|---|---|---|---|
| Yelp Review Polarity | 2 | 156 | 560,000 | 38,000 | Sentiment Analysis |
| Yelp Review Full | 5 | 158 | 650,000 | 50,000 | |
| Amazon Review Polarity | 2 | 91 | 3,000,000 | 650,000 | |
| Amazon Review Full | 5 | 93 | 3,600,000 | 400,000 | |
| AG's News | 4 | 44 | 120,000 | 7,600 | News Classification |
| Sogou News | 5 | 579 | 450,000 | 60,000 | |
| Yahoo! Answers | 10 | 112 | 1,400,000 | 60,000 | Question Answer |
| DBPedia | 14 | 55 | 560,000 | 70,000 | Ontology Extraction |

## 4.3 IMPLEMENTATION DETAILS

For data preprocessing, all the texts of datasets are tokenized by Stanford tokenizer and all words are converted to lower case. Words that appear only in one document are treated as out-of-vocabulary (OOV) items, and all stop words as well as symbols are kept. Additionally, length of $c$ padding are added to both the head and tail of each document.

For our models, optimal hyperparameters are tuned with 10% of the training set on Yelp Review Full dataset, and identical hyperparameters are applied to all datasets: the dimension of word embedding is 128, the region size is 7 which means the shape of local context unit matrix of each word is $128 \times 7$, the initial learning rate is set to $1 \times 10^{-4}$, and the batch size is 16. For optimization, the embeddings of words and the units are randomly initialized with Gaussian Distribution. Adam (Kingma & Ba, 2014) is used as the optimizer. We do not use any extra regularization methods, like L2 normalization or dropout. Algorithms are entirely implemented with TensorFlow and trained on NVIDIA Tesla P40 GPUs. The code [1] is publicly available on the Internet.

## 4.4 RESULTS

Table 2: Test Set Accuracy [%] Compared to other Methods on several Datasets

| Model | Yelp P. | Yelp F. | Amz. P. | Amz. F. | AG | Sogou | Yah. A. | DBP |
|---|---|---|---|---|---|---|---|---|
| BoW | 92.2 | 58.0 | 90.4 | 54.6 | 88.8 | 92.9 | 68.9 | 96.6 |
| ngrams | 95.6 | 56.3 | 92.0 | 54.3 | 92.0 | 97.1 | 68.5 | 98.6 |
| ngrams TFIDF | 95.4 | 54.8 | 91.5 | 52.4 | 92.4 | 97.2 | 68.5 | 98.7 |
| char-CNN | 94.7 | 62.0 | 94.5 | 59.6 | 87.2 | 95.1 | 71.2 | 98.3 |
| char-CRNN | 94.5 | 61.8 | 94.1 | 59.2 | 91.4 | 95.2 | 71.7 | 98.6 |
| bigram-FastText | 95.7 | 63.9 | 94.6 | 60.2 | 92.5 | 96.8 | 72.3 | 98.6 |
| VDCNN | 95.7 | 64.7 | **95.7** | **63.0** | 91.3 | 96.8 | 73.4 | 98.7 |
| D-LSTM | 92.6 | 59.6 | - | - | 92.1 | 94.9 | 73.7 | 98.7 |
| W.C.region.emb | **96.4** | **64.9** | 95.1 | 60.9 | **92.8** | 97.6 | **73.7** | **98.9** |
| C.W.region.emb | 96.2 | 64.5 | 95.3 | 60.8 | **92.8** | 97.3 | 73.4 | **98.9** |

Table 2 is the summary of the experimental results. We use underscores to represent the best published results, and bold the best records. On six datasets of eight, our models beat or match the state-of-the-art with a performance gain highest to 0.7%. We beat all the previous models on all datasets except VDCNN, while the latter performs almost best on all classification tasks before. As a result, we slightly win VDCNN on six datasets and lost in two of Amazon datasets. Detailed experimental results including best performance epoch and training time for all listed datasets are reported in Appendix A.

---

[1] `https://github.com/text-representation/local-context-unit`

Furthermore, the upper layer structure of our models only uses a summing up operation, which is more concise and robust than any other deep or complex models. In fact, both of our two proposed models are effective against previous models.

### 4.5 EXPLORATORY EXPERIMENTS

In this subsection, we are going to do a set of exploratory experiments to study the effect of each component of our model. Typical cases will be analyzed to validate properties of various aspects of our models. Considering the limitation of paper space, we only analyzed the Word-Context region embedding model in our exploratory experiments.

#### 4.5.1 EFFECT OF REGION SIZE AND EMBEDDING SIZE

Our method uses a fixed size of region as contextual information just like CNN. So the selection of region size really matters. A small region may lose some long distance patterns, whereas large regions will bring into more noises. Luckily, our models seem to be fairly insensitive towards kinds of datasets. Actually, we just use identical region size 7 for all datasets and it is able to outperform the best published results ever.

Figure 2a describes the performance on Yelp Review Full with different region sizes, and when the size equals to 1, the result is quite close to unigram FastText(accuracy 60.7%), but still gets a 0.6% promotion. Intuitively, the middle word cannot influence other words except itself when the size equals to 1. The performance increases with the growth of region size up to 7.

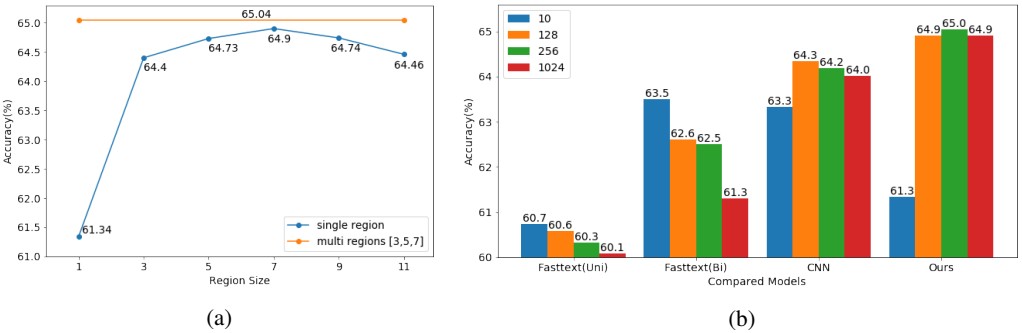

(a)               (b)

Figure 2: Effect of the hyperparameters ($region\ size$ and $embedding\ size$) on Yelp Review Full dataset. (a) shows the comparison of single fixed region size 7 and multi sizes combination [3,5,7] and (b) shows the effect of different settings of embedding size among four kinds of models, unigram FastText, bigram FastText, CNN and ours. We use region size 7 for CNN and ours.

Furthermore, we experiment our models with the combination of multi region sizes. Here we use the approach of sharing context units among each region, where the parameters of local context units of smaller regions are just the slice of the longest one. Region embeddings of different sizes are concatenated for final classification. In figure 2a, the combination of multi region sizes *3,5,7* is slightly better than the best single region size 7. The effectiveness of multi-size combination can be explained by the difference of influence ranges between words. For example, in sentiment analysis, word *very* only emphasizes the next word while *however* may lay stress on a wide range of the following words.

In addition to the analysis of region sizes, we further study the influence of word embedding dimensions. Figure 2b lists the comparative results on Yelp Review Full with different embedding dimensions. The result shows that our model is more robust to overfitting than FastText and CNN with the word embedding dimension increasing. In fact, the amount of parameters in our models is relatively large. Since we learn a specific unit for each word, under the same word embedding dimension, our parameter size has been expanded by *region size* times, the parameters number is $v \times h + v \times (2 \times c + 1) \times h + h \times m + m$, where $m$ is the number of classes. Specific numbers parameters for different region sizes are listed in Appendix A. Notice that the sizes of parameters are relatively consistent among *1024 in FastText*, *1024 in CNN* and *128 in ours*.

### 4.5.2 EFFECT OF CONTEXT UNIT

In this section, we explore some comparative experiments to show the effectiveness of our proposed word specific context unit. The experiments are employed based on unigram FastText baseline, which has similar upper layer structure with our models. Table 3 illustrates the results.

Table 3: Comparative decomposition results on Yelp Review Full dataset. For FastText(Unigram), embedding dimension is 10. For FastText(Win-pool), W.C.region.emb(Scalar) and W.C.region.emb(our model), region size is 7 and embedding dimension is 128
.

| Decomposition | Performance(%) |
|---|---|
| FastText(Unigram) | 60.73 |
| FastText(Win-pool) | 61.01(+**0.28**) |
| W.C.region.emb(Scalar) | 63.18(+**2.45**) |
| W.C.region.emb(Our model) | 64.9(+**4.17**) |

Firstly, we remove the entire context units from our model, which means it is just a variant version of unigram FastText, we call it FastText(Win-pool). The difference is that FastText sums up the word embeddings directly while FastText(Win-pool) sums up the window pooled embeddings in a stride of 1. It yields a slightly accuracy gain of 0.28% than unigram FastText.

Secondly, we apply a simplified scalar version of context units to FastText(Win-pool). Distinguishable, the context unit of each word has the shape with $1 \times (2 \times c + 1)$, hence it can be regarded as a broadcasting operation on corresponding word embeddings of its local context. We name this method W.C.region.emb(Scalar). Compared to the non-scalar method, it yields a huge parameter size reduction, but it already yields a significant gain of 2.45%.

Furthermore, W.C.region.emb(our model) is the variant version of W.C.region.emb(Scalar) where each column of scalar context unit is expanded to a dense vector. Each word's context unit has a shape with $h \times (2 \times c + 1)$. Adding the low dimensional dense context unit improves the performance by 4.17%. We can sense much from the procedure of decomposition, with the help of context unit, even a simpler scalar version promotes a lot.

To have a better understanding of what context unit actually capture, heat maps are plotted for chosen word samples. Representative adversarial conjunctions like *however*, *but*, modifiers like *very*, *good*, *bad* and nouns like *food*, *morning* are listed in Figure 3.

For each row of the figure, the intensity of the surrounding color box reflects the emphasis degree in the view of the middle word. Qualitative but not fully rigorous, a normalized L2-norm of each column in context unit is used to render the shade. Region size 7 is adopted default, annotation $l_i(i \leq 3)$ is denoted as left columns of the specific context unit, while $r_i(i \leq 3)$ denoted as the right part.

What the figure reflects are consistent with intuitive priors of human beings. In the perspective of *however*, right contexts play the key role for classification polarity because of the emotional reversal, the color is indeed deeper in $r_i$ than $l_i$, so does *but*. For word *very*, $r_1$ is more prominent than the rest of all, which captures some modified patterns like *very happy* or *very sad*. For word *good*, tendencies will be completely different for patterns like *not good*, *very good* and *not that good*, which are intensive negative, intensive positive and slightly hesitated, separately, the position of $l_1$ will be strengthened as a result, so does word *bad*. There are significant differences between two nouns *food* and *morning*. The heat map of word *food* implies patterns like *delicious food* or *food was mediocre*, while the word *morning* has fewer valuable patterns for classification.

Actually, from the motivation of word specified context units, we would like to believe this feature helps capture syntactic and semantic influences of words on surrounding words at relative positions.

### 4.5.3 VISUALIZATION

In this subsection, we will try to visualize the contribution of each word and selected phrase to classification. Detailed visualization techniques have been introduced in Li et al. (2015). Here we generalize it to the color rendering of multi-category version. Notice that for our model, not the

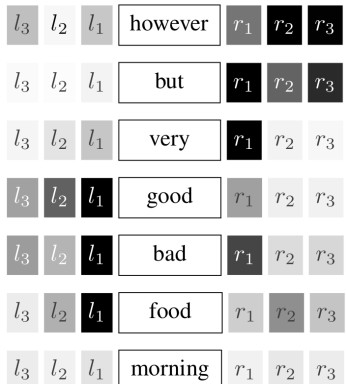

Figure 3: Heat maps of chosen words trained on Yelp Review Polarity, which is a binary sentiment analysis dataset. Each row represent the context unit of the middle word. Region size is 7 and embedding size is 128.

original embedding acts here, but the accumulation of the projected embedding of each word on its surrounding words.

Table 4: Visualization of chosen samples on Yelp Review Polarity dataset. Green denotes positive contribution while red denotes negative. Two methods are compared without context unit(No C-unit) and with context unit(With C-unit).

| Method | Sentence Samples | | | | | | | | | | Phrase |
|---|---|---|---|---|---|---|---|---|---|---|---|
| No C-unit | get | your | wallet | ready | , | the | prices | are | crazy | high | prices are crazy high |
| With C-unit | get | your | wallet | ready | , | the | prices | are | crazy | high | prices are crazy high |
| No C-unit | | nothing | remarkable | , | but | not | bad | either | | | but not bad either |
| With C-unit | | nothing | remarkable | , | but | not | bad | either | | | but not bad either |

For clarity, we choose a binary classification task of sentiment analysis. In Table 4, we list two cases in Yelp Review Polarity dataset, in which our model behaves as expected. Words and artificially selected phrases are highlighted green if they are positive factors, red if they are negative. The intensity of the color indicates the degree of the polarity.

To have a better comparison, the results of with and without context unit methods are both visualized. We abbreviate them as *With C-unit* and *No C-unit*, respectively. For sentence *get your wallet ready, the prices are crazy high*, if no context unit is adopted, the word color reflects its word embedding, which is context-free. The polarity of *crazy* is positive, and *high* is negative. Because the intensity of *crazy* is higher than *high*, the polarity of phrase *prices are crazy high* is totally positive, which is a mistake. But with context unit, things have changed quite a bit, the polarities of words are context dependent. Under the influence of *high*, the positive polarity of *crazy* vanishes and phrase *prices are crazy high* performs negative overall. For another case *nothing remarkable, but not bad either*, things seem more interesting. Without context-unit , *remarkable* is positive, while *nothing, not, bad* perform negative, respectively. But with context unit, the polarity of the part ahead of *but* weakens, meanwhile the polarities of *not* and *bad* flips. As a result, phrase *but not bad either* performs positive overall.

## 5 CONCLUSION

This paper proposed two novel architectures for text classification tasks, which learn task specific region embeddings without hand crafted features. To utilize the word specific influences of each word on its context words, a local context unit for each word is learned in addition to word embedding. Our models achieve state-of-the-art performances on six benchmark text classification datasets, and

the visualization experiments show that our proposed local context unit can capture the semantic and syntactic information for each word.

Noticed the power of the local context unit on learning task related region embeddings, we are interested in its ability to unsupervised and semi-supervised learning. At the same time, we are also curious about whether we can achieve better results by introducing more complex upper layers on text classification, and other natural language processing tasks.

## ACKNOWLEDGMENTS

This paper is supported by National Basic Research Program of China (973 program No.2014CB340505). We gratefully thank the anonymous reviewers for their insightful comments.

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

# APPENDIX

## A  DETAILED EXPERIMENTAL RESULTS

To have a further insight about our reported results, we list the training time and best testing performance epoch for kinds of region sizes(*5,7,9*) in detail. For all the 8 datasets in Table 5, hyperparameters are kept in line with section 4.3. The dimension of word embedding is 128, the region size is 7, the initial learning rate is set to $1 \times 10^{-4}$, and the batch size is 16. Epoch index starts from 0 and training time is reported per epoch. We choose the model of word-context region embedding here.

Table 5: Experimental detailed records on several datasets

| Dataset | Vocabulary Size | W.C Region Size | Parameters Number | Best Epoch | Training Time(Mins) | Accuracy(%) |
|---------|----------------|-----------------|-------------------|------------|---------------------|-------------|
| Yelp P. | 115298 | 5 | 88,549,122 | 3 | 27 | 96.34 |
|         |        | 7 | 118,065,410 | 3 | 36 | 96.39 |
|         |        | 9 | 147,581,698 | 3 | 43 | 96.38 |
| Yelp F. | 124273 | 5 | 95,442,309 | 3 | 34 | 64.73 |
|         |        | 7 | 127,256,197 | 2 | 43 | 64.90 |
|         |        | 9 | 159,070,085 | 2 | 52 | 64.74 |
| Amz. P. | 394385 | 5 | 302,887,938 | 2 | 336 | 95.07 |
|         |        | 7 | 403,850,498 | 2 | 402 | 95.23 |
|         |        | 9 | 504,813,058 | 2 | 589 | 95.06 |
| Amz. F. | 356312 | 5 | 273,648,261 | 1 | 300 | 60.83 |
|         |        | 7 | 364,864,133 | 1 | 395 | 60.93 |
|         |        | 9 | 456,080,005 | 1 | 490 | 61.05 |
| AG | 42783 | 5 | 32,857,860 | 6 | 2 | 92.81 |
|    |       | 7 | 43,810,308 | 4 | 3 | 92.89 |
|    |       | 9 | 54,762,756 | 5 | 4 | 92.82 |
| Sogou | 99394 | 5 | 76,335,237 | 7 | 27 | 97.6 |
|       |       | 7 | 101,780,101 | 9 | 33 | 97.63 |
|       |       | 9 | 127,224,965 | 10 | 39 | 97.56 |
| Yah.A. | 361926 | 5 | 277,960,458 | 1 | 160 | 73.42 |
|        |        | 7 | 370,613,514 | 1 | 210 | 73.66 |
|        |        | 9 | 463,266,570 | 2 | 256 | 73.68 |
| DBP | 227863 | 5 | 175,000,590 | 3 | 37 | 98.87 |
|     |        | 7 | 233,333,518 | 2 | 48 | 98.89 |
|     |        | 9 | 291,666,446 | 3 | 60 | 98.94 |

We also report results of several repeated runs in Table 6 to exclude the effect of randomness and ensure reproducibility. Five independent runs are conducted on each dataset of Yelp.P and Yelp.F, where both performance variances are within 0.11% on accuracy.

Table 6: Performance variances through several repeated runs on Yelp Datasets

| Dataset | Tries Num. | W.C region size | Best Epoch | Accuracy(%) | Performance Variance |
|---------|-----------|-----------------|------------|-------------|----------------------|
| Yelp P. | 0 | 7 | 3 | 96.39 | |
|         | 1 | 7 | 4 | 96.36 | |
|         | 2 | 7 | 4 | 96.41 | $\leq \%0.11$ |
|         | 3 | 7 | 3 | 96.38 | |
|         | 4 | 7 | 2 | 96.46 | |
| Yelp F. | 0 | 7 | 2 | 64.90 | |
|         | 1 | 7 | 2 | 64.94 | |
|         | 2 | 7 | 1 | 64.87 | $\leq \%0.11$ |
|         | 3 | 7 | 1 | 64.86 | |
|         | 4 | 7 | 2 | 64.98 | |

