# OpenReview forum: "A New Method of Region Embedding for Text Classification"
_ICLR.cc/2018/Conference — Accept (Poster)_

### Official Review · AnonReviewer1 · 2017-11-25
**Review of "Bag of region embeddings ..."**

**Rating:** 6
**Confidence:** 4

**Review:**

The authors propose a mechanism for learning task-specific region embeddings for use in text classification. Specifically, this comprises a standard word embedding an accompanying local context embedding.

The key idea here is the introduction of a (h x c x v) tensor K, where h is the embedding dim (same as the word embedding size), c is a fixed window size around a target word, and v is the vocabulary size. Each word in v is then associated with an (h x c) matrix that is meant to encode how it affects nearby words, in particular this may be viewed as parameterizing a projection to be applied to surrounding word embeddings. The authors propose two specific variants of this approach, which combine the K matrix and constituent word embeddings (in a given region) in different ways. Region embeddings are then composed (summed) and fed through a standard model.

Strong points
---
+ The proposed approach is simple and largely intuitive: essentially the context matrix allows word-specific contextualization. Further, the work is clearly presented.

+ At the very least the model does seem comparable in performance to various recent methods (as per Table 2), however as noted below the gains are marginal and I have some questions on the setup.

+ The authors perform ablation experiments, which are always nice to see.

Weak points
---
- I have a critical question for clarification in the experiments. The authors write 'Optimal hyperparameters are tuned with 10% of the training set on Yelp Review Full dataset, and identical hyperparameters are applied to all datasets' -- is this true for *all* models, or only the proposed approach?

- The gains here appear to be consistent, but they seem marginal. The biggest gain achieved over all datasets is apparently .7, and most of the time the model very narrowly performs better (.2-.4 range). Moreoever, it is not clear if these results are averaged over multiple runs of SGD or not (variation due to initialization and stochastic estimation can account for up to 1 point in variance -- see "A sensitivity analysis of (and practitioners guide to) CNNs..." Zhang and Wallace, 2015.)

- The related work section seems light. For instance, there is no discussion at all of LSTMs and their application to text classificatio (e.g., Tang et al., EMNLP 2015) -- although it is noted that the authors do compare against D-LSTM,  or char-level CNNs for the same (see Zhang et al., NIPs 2015). Other relevant work not discussed includes Iyyer et al. (ACL 2015). In their respective ways, these papers address some of the same issues the authors consider here.

- The two approaches to inducing the final region embedding (word-context and then context-word in sections 3.2 and 3.3, respectively) feel a bit ad-hoc. I would have appreciated more intuition behind these approaches.

Small comments
---
There is a typo in Figure 4 -- "Howerver" should be "However"

*** Update after author response ***

Thanks to the authors for their responses. My score is unchanged.

---

> ### Author Response · Authors · 2018-01-03
> **Response to ICLR 2018 Conference Paper65 AnonReviewer1**
>
> Thanks for your valuable comments. We will explain your concerns point by point:
>
> 1)"The authors write 'Optimal hyperparameters are tuned with 10% of the training set on Yelp Review Full dataset, and identical hyperparameters are applied to all datasets' -- is this true for *all* models, or only the proposed approach? "
>
> This is true  only for the proposed approach. Results of the previous methods are their best results reported in corresponding previous papers, in which the hyperparameters are not identical for each datasets. (reference section 3.3 in D-LSTM, Table 5 in VDCNN, Table 1 and second paragraph of section 3.1 in FastText, section 4.2 in char-CRNN).
>
>
> 2) "The gains here appear to be consistent, but they seem marginal. The biggest gain achieved over all datasets is apparently .7, and most of the time the model very narrowly performs better (.2-.4 range). Moreover, it is not clear if these results are averaged over multiple runs of SGD or not."
>
> The result are averaged over multiple runs. We have experimented the performance variance in independent tries on yelp datasets, the results are reported in Appendix. A.
>
> In this paper, we want to show that, with the ability of word-specific contextualization given by our proposed local context unit, our simple model can consistently beats or achieves the state-of-the-art results on almost all text classification tasks against to previous methods (traditional and deep models). This gives us an insight to represent and understand natural language by word specific context units in our future work. Therefore, we didn't use any trival tricks (e.g. multi-region-size which have been proved can improve the performance) and extra regularization methods. In fact, the gains are not so marginal since the best previous method's gains are similar and even less than ours on some datasets.
>
> 3)The related work section seems light.
>
> We have improved and completed related work section now.
>
> 4)The two approaches to inducing the final region embedding (word-context and then context-word in sections 3.2 and 3.3, respectively) feel a bit ad-hoc. I would have appreciated more intuition behind these approaches.
>
> The main intuition we compose the region embeddings in two approaches is the flowing: We consider the semantic of a given region is derived from the mutual influences of the words in this region. Since the regions can be regarded as snapshots of a window sliding on a document, whose middle words are contiguous,  we can just focus on the middle word influences on the context words, or the context words' influences on the middle word. According to the property of local context unit introduced in section 3.1, embeddings and units are used in two ways to address these influences, respectively. Finally, to extract the most predictive information and then produce a fixed length vector representation, a max pooling operation is used.
>
> We also revised sections 3.2 and 3.3 of this paper.

---

### Official Review · AnonReviewer3 · 2017-11-26
**New model for text classification**

**Rating:** 6
**Confidence:** 5

**Review:**

The authors present a model for text classification. The parameters of the model are an embedding for each word and a local context unit. The local context unit can be seen as a filter for a convolutional layer, but which filter is used at location i depends on the word at location i (i.e. there is one filter per vocabulary word). After the filter is applied to the embeddings and after max pooling, the word-context region embeddings are summed and fed into a neural network for the classification task. The embeddings, the context units and the neural net parameters are trained jointly on a supervised text classification task. The authors also offer an alternative model, which changes the role of the embedding an the context unit, and results in context-word region embeddings. Here the embedding of word i is combined with the elements of the context units of words in the context. To get the region embeddings both model (word-context and context-word) combine attributes of the words (embeddings) with how their attributes should be emphasized or deemphasized based on nearby words (local context units and max pooling) while taking into account the relative position of the words in the context (columns of the context units).

The method beats existing methods for text classification including d-LSTMs , BoWs, and ngram TFIDFs on held out classification accuracy. the choice of baselines is convincing. What is the performance of the proposed method if the embeddings are initialized to pretrained word embeddings and a) trained for the classification task together with randomly initialized context units b) frozen to pretrained embeddings and only the context units are trained for the classification task?

The introduction was fine. Until page 3 the authors refer to the context units a couple of times without giving some simple explanation of what it could be. A simple explanation in the introduction would improve the writing.
The related work section only makes sense *after* there is at least a minimal explanation of what the local context units do. A simple explanation of the method, for example in the introduction, would then make the connections to CNNs more clear. Also, in the related work, the authors could include more citations (e.g. the d-LSTM and the CNN based methods from Table 2) and explain the qualitative differences between their method and existing ones.

The authors should consider adding equation numbers. The equation on the bottom of page 3 is fine, but the expressions in 3.2 and 3.3 are weird. A more concise explanation of the context-word region embeddings and the word-context region embeddings would be to instead give the equation for r_{i,c}.

The included baselines are extensive and the proposed method outperforms existing methods on most datasets. In section 4.5 the authors analyze region and embedding size, which are good analyses to include in the paper. Figure 2 and 3 could be next to each other to save space.
I found the idea of multi region sizes interesting, but no description is given on how exactly they are combined. Since it works so well, maybe it could be promoted into the method section? Also, for each data set, which region size worked best?

Qualitative analysis: It would have been nice to see some analysis of whether the learned embeddings capture semantic similarities, both at the embedding level and at the region level. It would also be interesting to investigate the columns of the context units, with different columns somehow capturing the importance of relative position. Are there some words for which all columns are similar meaning that their position is less relevant in how they affect nearby words? And then for other words with variation along the columns of the context units, do their context units modulate the embedding more when they are closer or further away?

Pros:
 + simple model
 + strong quantitative results

Cons:
 - notation (i.e. precise definition of r_{i,c})
 - qualitative analysis could be extended
 - writing could be improved

---

> ### Author Response · Authors · 2018-01-03
> **Response to ICLR 2018 Conference Paper65 AnonReviewer3**
>
> Thank for your suggestions. We will explain your concerns point by point.
>
> 1) "What is the performance of the proposed method if the embeddings are initialized to pretrained word embeddings and a) trained for the classification task together with randomly initialized context units b) frozen to pretrained embeddings and only the context units are trained for the classification task?"
> We evaluated the experiments with pretrained word embeddings on Yelp.F and Yelp.P dataset:
>
> *Datasets*        *Method*        Best epoch(start from 0)*        *Accuracy*
> Yelp F.        Random        1        0.649500
> Yelp F.        Finetune        1        0.638580
> Yelp F.        Frozen        2        0.633060
> Yelp P.        Random        2        0.963895
> Yelp P.        Finetune        1        0.962500
> Yelp P.        Frozen        2        0.960842
>
> The word embeddings are pre-trained by Wikipedia+Gigaword 5 glove with 200 dimensions, region size is 7.  The result of a) is similar with randomly initialized word embeddings, and result of b) is slightly worse.
> Intuitively, pre-trained word embeddings should have a role, but maybe should not be applied directly. In fact, we will explore the way to apply local context unit to semi-supervised and unsupervised learning in our future work.
>
> 2) "Until page 3 the authors refer to the context units a couple of times without giving some simple explanation of what it could be. A simple explanation in the introduction would improve the writing." && "the authors could include more citations (e.g. the d-LSTM and the CNN based methods from Table 2) and explain the qualitative differences between their method and existing ones."
>
> We have deeply rewrited the introduction section and added citations related to ours.
>
> 3)"The authors should consider adding equation numbers. The equation on the bottom of page 3 is fine, but the expressions in 3.2 and 3.3 are weird. A more concise explanation of the context-word region embeddings and the word-context region embeddings would be to instead give the equation for r_{i,c}. "
>
> Thanks for your suggestions, equation numbers have been added, and expressions in 3.2 and 3.3 have been updated. We also add explanations and discussions in 3.2 to make this paper clearer.
>
> 4)"I found the idea of multi region sizes interesting, but no description is given on how exactly they are combined. Since it works so well, maybe it could be promoted into the method section? Also, for each data set, which region size worked best?"
>
> Detailed information about multi region sizes has been added at section 4.5.1. The gains of the multi region sizes method are not so large. As a natural extend for our core idea, we prefer to discuss it in the exploratory experiments sections. Performances for each data set with different region size have been reported in Appendix.A now.
>
> 5)"Are there some words for which all columns are similar meaning that their position is less relevant in how they affect nearby words? And then for other words with variation along the columns of the context units, do their context units modulate the embedding more when they are closer or further away? "
>
> We have discussed this issue at section 4.5.4 and added words whose positions are less relevant in how they affect nearby words, which is consistent with our previous hypothesis. As for the second question, we have evaluated the entire vocabulary from the perspective of statistics and no obvious differential distribution built on different columns. In fact, it seems size of half of region characters expression patterns more, instead of strong distance distinction.

---

### Official Review · AnonReviewer2 · 2017-11-28

**Rating:** 6
**Confidence:** 3

**Review:**

() Summary
In this paper, the authors introduced a new simple model for text classification, which obtains state of the art results on several benchmark. The main contribution of the paper is to propose a new technique to learn vector representation of fixed-size text regions of up to a few words. In addition to learning a vector for each word of the vocabulary, the authors propose to also learn a "context unit" of size d x K, where d is the embedding size and K the region size. Thus, the model also have a vector representation for pair of word and position in the region. Then, given a region of K words, its vector representation is obtained by taking the elementwise product of the "context unit" of the middle word and the matrix obtained by concatenating the K vectors of words appearing in the region (the authors also propose a second model where the role of word vectors and "context" vectors are exchanged). The max-pooling operation is then used to obtain a vector representation of size d. Then a linear classifier is applied on top of the sum of the region embeddings. The authors then compare their approach to previous work on the 8 datasets introduced by Zhang et al. (2015). They obtain state of the art results on most of the datasets. They also perform some analysis of their models, such as the influence of the region size, embedding size, or replacing the "context units" vector by a scalar. The authors also provide some visualisation of the parameters of their model.

() Discussion
Overall, I think that the proposed method is sound and well justified. The empirical evaluations, analysis and comparisons to existing methods are well executed. I liked the fact that the proposed model is very simple, yet very competitive compared to the state-of-the-art. I suspect that the model is also computationally efficient: can the authors report training time for different datasets? I think that it would make the paper stronger. One of the main limitations of the model, as stated by the authors, is its number of parameters. Could the authors also report these?

While the paper is fairly easy to read (because the method is simple and Figure 1 helps understanding the model), I think that copy editing is needed. Indeed, the papers contains many typos (I have listed a few), as well as ungrammatical sentences. I also think that a discussion of the "attention is all you need" paper by Vaswani et al. is needed, as both articles seem strongly related.

As a minor comment, I advise the authors to use a different letter for "word embeddings" and the "projected word embeddings" (equation at the bottom of page 3). It would also make the paper more clear.

() Pros / Cons:
+ simple yet powerful method for text classification
+ strong experimental results
+ ablation study / analysis of influence of parameters
- writing of the paper
- missing discussion to the "attention is all you need paper", which seems highly relevant

() Typos:
Page 1
"a support vectors machineS" -> "a support vector machine"
"performs good" -> "performs well"
"the n-grams was widely" -> "n-grams were widely"
"to apply large region size" -> "to apply to large region size"
"are trained separately" -> "do not share parameters"

Page 2
"convolutional neural networks(CNN)" -> "convolutional neural networks (CNN)"
"related works" -> "related work"
"effective in Wang and Manning" -> "effective by Wang and Manning"
"applied on text classification" -> "applied to text classification"
"shard(word independent)" -> "shard (word independent)"

Page 3
"can be treat" -> "can be treated"
"fixed length continues subsequence" -> "fixed length contiguous subsequence"
"w_i stands for the" -> "w_i standing for the"
"which both the unit" -> "where both the unit"
"in vocabulary" -> "in the vocabulary"

etc...

---

> ### Author Response · Authors · 2018-01-03
> **Response to ICLR 2018 Conference Paper65 AnonReviewer2**
>
> Thank you very much for suggestions and meticulous corrections for this paper. Your description of our work is accurate. We have addressed each of your comments:
>
> 1) Didn't report training time.
> We have reported the training time for each dataset with different region sizes in appendix A.
>
> 2) Didn't report number of parameters.
> Parameters have been discussed in 4.5.1, and reported in appendix A for different settings.
>
> 3) Typos and ungrammatical sentences in the paper.
> We have greatly improved the writing of this paper, including all the typos you pointed out and others textual errors. We will continue to improve the writing quality before the camera ready version.
>
> 4) The lack of discussion of the "attention is all you need" paper.
> This work has been discussed now in the related work section.
>
> 5) Should use different letters for "word embeddings" and the "projected word embeddings"
> We have improved the notations to make the paper clearer.

---

### Public Comment · (anonymous) · 2017-11-15
**Code of the proposed models**

In order to facilitate reviewers to reproduce our results, we share the implementation of our method at here(https://github.com/text-representation/local-context-unit).  We will formally open-source our code upon publishing the paper.

---

### Public Comment · ~Shagun_Sodhani1 · 2017-11-19
**Query about the datasets**

Hey authors

Thank you for sharing the implementation of the paper - it goes a long way towards ensuring reproducibility.

My understanding of the paper is the following: for each word in the vocabulary, along with learning a word vector, learn a context matrix. This context matrix would introduce a soft-attention kind of effect and is expected to be more powerful than the use of just a context vector for capturing the context. Please correct me if there is something wrong in my understanding :)

In the experiments section, the paper uses 8 different datasets. It would be helpful if the paper also mentions the average document length for the different datasets. That could be a crude proxy to understand how important is it to capture the long term dependencies in the document. Further, the simple baselines of BoW and ngrams give decent performance (around 90% for 5 datasets). Could it be the case that the datasets are not very "difficult"? I have not used these datasets and would be glad to know what do the authors feel about the same.

---

> ### Public Comment · (anonymous) · 2017-11-24
> **Further information of the datasets**
>
> Thank you for your comments and suggestions. We generally agree with your understanding of this paper， if there are any problems with the implementation, please let us know.
>
> About your questions with datasets:
>
> 1)The average document length for the different datasets are following, we will add them in the next revision:
>
>     *Dataset*       *Average Document Length*
>     Yelp P.         156.153767857
>     Yelp F.         157.641587692
>     Yah.A.          111.638106429
>     Sogou           578.654484444
>     DBP             55.3340017857
>     Amz.P.          90.8814119444
>     Amz.F.          92.7758653333
>     AG              43.6560583333
>
> It is worth to mention that the motivation of our method is not to capture the long distance dependence, but to capture the local features of the text from a new perspective(Although more complex upper layers like RNNs can be applied to capture the long term dependencies in the document, we simply used a bag of region embeddings upper layer).
>
>
> 2)The datasets are built from real world and are widely used to evaluate the model performance on text classification tasks(see details in the references of this paper). On some datasets such as AG and Sogou, the Bow and Bag-of-Ngrams can achieve 90%+ accuracy which indeed shows that these datasets are naturally easy to be separated, but we can see that our method achieves the state of the art results against the baselines and along with other deep models(LSTMs, CNNs), on some other more difficult datasets, such as multi-level sentiment analysis(Yelp.F., Amz.F), QA matching(Yah.A), our method yields a significant performance gain compared to the BoW and ngrams baselines (5%-8%), which shows that our method can capture local text features better.

---

### Comment · AnonReviewer3 · 2017-11-25
**Please clarify loss**

How are you training the local context units? Which loss are you optimizing to get the embeddings E and context parameters U?

---

> ### Public Comment · (anonymous) · 2017-11-26
> **They are jointly trained as classification model parameters with cross entropy loss**
>
> Thanks for your question.
>
> In this paper, we are going to produce the task related region embeddings which can be used to improve the performance of classification tasks, so the word embedding matrix E and local context unit matrix U are randomly initialized and jointly trained as classification model parameters, here we use the cross entropy loss as the classification loss.
>
> Noticed that the power of the local context unit on learning task related region embeddings, we are
> interested to explore its ability to semi-supervised or unsupervised learning in our future work, but in this paper, we only focus on developing a new mechanism to extract the predictive features on small text regions.

---

> > ### Comment · AnonReviewer3 · 2017-11-26
> > **What about the embeddings in competing methods?**
> >
> > Thank you so much for the clarifications! What about the embeddings of the competing methods, e.g. Fasttext? Are they pretrained? Are they retrained for the classification task at hand?

---

> > > ### Public Comment · (anonymous) · 2017-11-27
> > > **Embeddings in competing methods**
> > >
> > > Thank you for asking.
> > >
> > > In this paper, previous works' experimental results in Table 2 are reported from Joulin et al.(2016). From our knowledge, we believe FastText and the character based methods did not use pre-trained word embeddings, and we are not very sure whether the embeddings are pre-trained  in Discriminative LSTM from now on.  Performances on these eight datasets of word based CNN can be found in Zhang & LeCun (2015)(with and without pre-trained word embeddings), which show the effect of whether the embeddings are pre-trained.
> > >
> > > More details about those experiments under different conditions can be found in the original papers for each method.

---

### Public Comment · ~Jingyun_Liu1 · 2017-12-15
**Reproducibility  review**

This paper proposed a new technique to learn a vector representation of local context information, where the model learns a separate target embedding and context embedding for each word in a sequence. Under the proposed framework, there are two models: Context-Word and Word-Context. Word-Context uses the same context embedding for the target word on each context word, and Context-Word uses the same target embedding for each context word. Max pool is applied to get the final word representation. The authors evaluate the quality of the learned embeddings on text classification task where the word and context embeddings are directly feed into a linear classifier as input. The embedding for target word and context words and the neural net parameters are trained jointly on the supervised classification task.

We conduct experiments using both Word-Context and Context-Word on 4 out of the 8 datasets: Yelp Full Review, Yelp Review Polarity, AG news and DBP dataset. After running the experiments given the details provided, we compare our result against the paper’s claim, and conclude that the proposed model  is  mostly as effective as  the  paper  claimed (variance within 1% on accuracy), while in some cases we get results worse than the reported and we are not able to reproduce.

First, we reproduced the text classification tasks on the 4 datasets mentioned above. We find that there are small fluctuations (around 0.5% to 1%) on each result, but the results are in general consistent with the paper on all datasets except DBP and AG news. Surprisingly, our results on DBP dataset and AG News are not consistent with the authors’: our result is around 75% while the paper reports over 98% on DBP, and 88% while the paper reports 92% on AG news. We find no obvious reason for the model to fail, and thus the experiments on those dataset is not reproducible.

Next, we test several reported baselines on the full dataset. Due to time and computation limit, we reproduce 4 out of 8 baselines: Bag of Word (BoW), Bag of Ngrams, Bag of Ngrams with TFIDF and FastText.
The BoW, Ngram and Ngrams with TFIDF baselines are easy to implement with Scikit-learn library, but we find our results significantly lower than the reported results on an average of 5-7% in classification accuracy. The code for FastText is public online, and our reproduced results are in line with the claimed results for all datasets.

In addition, we conduct the experiments for analytical purpose as described in the paper. In the paper, the effect of region size and embedding size are studied separately, showing that a region size of 7 (3 words to the left of the target word and 3 words to the right) and an embedding of dimension 128 produces the best result. Here we also reproduce similar results on the same dataset following the instructions given in the paper. We find that the optimal region size is around 3-4 words on each side, and slight numerical difference (within 0.5% variation) might exist depending on the dataset and other hyperparameters such as embedding size. For example, on Word-Context model with embedding size 200 on Yelp Full Review, a region size of 9 turns out to yield the best results.

Furthermore, we conduct the experiments on the settings of different embedding size up until our time and computation power limit, and we find that the accuracy score is about 0.5% worse than the claim on large embedding sizes but about 1% better than the score reported on the paper.

Lastly, we examine the effect of context unit by comparing the performance of the proposed model with FastText baseline. As described in the paper, we train the model to learn a scalar representation of the word and its context, and compare it with the normal vector version and baselines. However, in practice it takes more than 2 days to train (on GPU) the model until convergence. Therefore we are not yet able to draw conclusion about the effectiveness of the model with scalar representation model compared to the one with vector form.

We also record the average cost for each model on each dataset. On large dataset such as Yelp Review, both proposed models take about 7-10 hours to converge with the reported hyperparameters, while on small datasets such as AG News, the models converge in 2-3 hours. The model learns relatively slowly compared to some baseline models such as FastText, which converges within half an hour. This is expected, given the number of parameters the proposed models have.

Overall, we conclude that the proposed methods are well justified and the empirical evaluations and analysis are well executed. Most results are reproducible within certain time and computation limit, but there also exist certain experiments that are not reproducible.

---

> ### Author Response · Authors · 2017-12-17
> **Reply**
>
> Thank you very much for the reproducing experiments and suggestions about this paper. We have updated our code and discussed the common issues about the reproducibility.
> In summary, within 1% variance gap can be explained by the 90% training data in the published configure as default, while we use 100% in the paper results.
> The problem of slow convergence may be caused by the learning rate. In the example configure it was 1e-5 which declared 1e-4 in the paper.
> The significant different reproducing results on DBPedia and AG News can be explained by the preprocess bug in our published code, and we have fixed it.
> Please see our latest comments to get more detailed information.
>
> We have add scalar mode context unit in our code, which should not take 2 days long to train the model until convergence, could you refer our implementation or share yours so we can find out the problem?

---

### Public Comment · (anonymous) · 2017-12-16

This paper proposed a new method on document classification task. In addition to word embedding, it assigns a context unit, which is a matrix, to each word. Given a region, the method uses these matrix’s columns and the word embedding vectors within the region to compute element-wise multiplication. The results are some projected vectors. Then they apply element-wise max-pooling for these projected vectors. In the end, the model extracts a feature vector on each N consecutive words and then it computes the average of these vectors (or computes the weighted sum). This final vector can be regarded as the feature vector of the entire document. Finally, the model feeds the vector into a fully connected layer and gets the classification results. This feature construction process is similar to that of the N-gram model since they all extract a feature vector from each region of fixed length, except that, in N-gram, each consecutive N words is used as a feature directly.
The authors justified the effectiveness of their model by comparing the test set accuracy to several well-known baseline models on 8 different data sets. Based on their result, the model outperformed all previous methods on all data sets except for VDCNN on two Amazon data sets. In general, the model achieved an overall outstanding performance in terms of accuracy. The paper also carried out several exploratory experiments to study the effect of each hyper-parameter on the model. The authors compared the effectiveness of the model under various region sizes and embedding sizes. They concluded from the results that their model is robust to the over-fitting problem. The paper also investigated the effects of context unit by successively removing each independent method and comparing the performance of each resulting model. It showed that accuracy increased every time when they added one more component to the model. Furthermore, a significant improvement occurred when the context unit was fully utilized.

Our result:
For Yelp Review Polarity data set, our test set accuracy does not match the reported accuracy, yet they are consistent to some extent. Despite the subtle disagreement, our result still exceeds the all test accuracy form previous work by at least 0.6\%. When comparing the results on Yelp Review Full data set, however, we observe large discrepancy. The authors claimed that their model achieved a test set accuracy of 64.9% and as such beat all previous models whereas our experiment yields an accuracy of 64.6% which is a bit far from so-claimed 64.9%. Moreover, based on our result, the proposed model also fails to outperform VDCNN.
We notice that, for all region sizes, we obtained a test set accuracy lower than the one reported by the authors. Based on our result, Best performance is achieved when region size is set to be 9 as opposed to the paper's result that the performance continues to increase with the growth of region size up to 7. Nevertheless, as the region size increases, model performance from two results firstly get dramatically promoted and begin to decline after some point.

Pros:
-The paper is easy to understand. Moreover, the authors used a graph to illustrate their model structure which is helpful.
-The model and the idea are simple, which makes the model easy to interpret.
-Code is provided which makes it easy to reproduce the result.
-The paper provided results and comparisons on several benchmarks with several state-of-the-art models.
Cons:
-Computational time for each task was not showed in this paper. If the authors provided the running time with different choice of hyper-parameters, it will be helpful for the readers to choose the combination of hyperparameters so as to best fit their computation environment when time is limited.
-The hyper-parameter settings used for producing results in the paper were not clearly stated. For example, in table 2 of this paper the epoch size was not given. There is ambiguity of all the numerical results. The accuracy can be either interpreted as the average accuracy of several repeated runs of this task or interpreted as the highest accuracy obtained from running the same task repeatedly for several times.
-The open source code has one major issue where the stop condition in trainer.py/function train() will result the whole training process to be forcefully terminated with error before computing the test and dev data accuracy of last epoch of the training process.
-The naming of variables in this paper is inconsistent and does not align with the open source code variable name, such as, hyper-parameter "region size" are renamed as "window size" in section 4.5.2.

---

> ### Author Response · Authors · 2017-12-17
> **Reply**
>
> Thank you very much for the reproducing experiments and suggestions about this paper. We have updated our code and discussed the common issues about the reproducibility.
> In summary, within 1% variance gap can be explained by the 90% training data in the published configure as default, while we use 100% in the paper results.
> Please see our latest comments to get more detailed information.
>
> Thank you for pointing out, we have fix the stop condition issue in train.py and refined the code.

---

### Public Comment · (anonymous) · 2017-12-16
**Reproducibility review**

New Proposed method of Text Classification
Summary
Within this paper, the author proposed two fresh text classification methods that learn task specific region embeddings without hand crafted features. In the model, there are two attributes for every word, which are generally word embeddings representing regions and local context unit which interacts with the word's context. After learning the bag of region embeddings which is either word-context embedding or context-word embedding, a linear classifier is used for classification. The authors implemented the context unit model and compared them with other 8 baseline methods using 8 datasets and showed the beating results over all previous models on all datasets except VDCNN on Amazon datasets. Besides, they explored the effects of a hyper-parameter of selecting region size since small region size loses patterns of long distances and large region size gets more noises. Additionally, they experimented with the effect of context window size and the embedding size. Finally, they visualized the contribution of each word and selected phrase of classification.
Discussion:
Through repetitive validations on baseline methods as well as proposed methods by the authors, we proved some baseline results are reproducible, however, some are not. To be more specific, we derived comparable results to the ones referenced by the paper using FastText Bigram or Unigram method. Nevertheless, when we tried to fit Dbpedia dataset with Bag of Words or Ngrams-TFIDF model, the prediction accuracy was low, achieving only around 70% where the authors claimed it would reach 96.6%. Thus, we suggest the authors to manually implement several baselines to make sure the reliability of the source data. After the reproduction of different embedding sizes on FastText Uni. & Bigram, the prediction accuracy is not declining as the embedding size increases. The advantage of using proposed method instead of FastText to avoid over-fitting is not obvious.
For the experiments using local context unit models, we reproduced the baseline run with optimal parameters on four datasets and studied the effects of embedding sizes and window sizes. We obtained similar results on Yahoo! Answers and Yelp Review Full datasets, with 64.6% and 68.9% respectively. When training with Word Context model, the best results we obtained on DBPedia within 20 epochs is 71.7%, differing significantly from the claimed results 98.9%. We carefully checked the source code and performed multiple trainings with different random initialization, however the accuracy didn’t improve. We doubt whether this problem is cause by implementation issue or the discrepancies in terms of training parameters. Moreover, we obtained 87% accuracy on AG’s News within 39 epochs, 5% less than the claimed results.
Due to the limited the computing power, we validate the effect of window sizes we obtained the results with only 8 ~ 12 epochs, except for windows size 1 and 7 of more than 20 epochs. However, the accuracy increase rate drops between adjacent epochs by the end of training on each dataset. Also, a clear distribution that is similar to the one illustrated in the paper is observed: 61.01%, 63.00%, 63.92%, 64.6%, 63.15%. We can thus conclude that the variation across window sizes is reproducible. However, the mixed region sizes are not reproduced because the implementation of this is not specified by neither the paper nor the source code.
We failed to perform FastText (Win-pool) and W.C. region embedding with one dimensional context unit because of some implementation issues which was hard to overcome. We suggest authors releasing the implementation for the variation or more verbose guild-lines for customizing the code.
Strength:
+ Proposed model is very intuitive, the representation of the architecture of C.M. & M.C. region embedding is clear and easy to follow.
+ Exploratory experiments on embedding size effect and region effect as well as the visualization make it more vivid
Weakness:
- The authors didn’t manually implement the baseline methods through 8 datasets.

---

> ### Author Response · Authors · 2017-12-17
> **Reply**
>
> Thank you very much for the reproducing experiments and suggestions about this paper. We have updated our code and discussed the common issues about the reproducibility.
> In summary, within 1% variance gap can be explained by the 90% training data in the published configure as default, while we use 100% in the paper results.
> The problem of slow convergence may be caused by the learning rate. In the example configure it was 1e-5 which declared 1e-4 in the paper.
> The significant different reproducing results on DBPedia and AG News can be explained by the preprocess bug in our published code, and we have fixed it.
> Please see our latest comments to get more detailed information.
>
> Could you please share which datasets you applied on FastText Uni. & Bigram with different embedding sizes? Since some datasets like DBPedia were preprocessed incorrectly, experiments on these datasets may lead different conclusion.
>
> Interestingly，we found there is a hidden part in the .tex file (%experiment notes part)of the original FastText paper(https://arxiv.org/abs/1607.01759, click other formats link). Embedding size 10 is better than 100 for both unigram&bigram in Fasttext.
>
> Model && AG & Sogou & DBP & Yelp P. & Yelp F. & Yah. A. & Amz. F. & Amz. P. \\
> %Ours, $h=100$                         && 91.0 & 92.6 & 98.2 & 92.9 & 59.6 & 70.7 & 55.3 & 90.9 \\
> %Ours, $h=100$, bigram                 && 92.4 & 96.4 & 98.5 & 95.7 & 63.7 & 71.9 & 59.2 & 94.5 \\
> \texttt{fastText}, $h=10$             && 91.5 & 93.9 & 98.1 & 93.8 & 60.4 & 72.0 & 55.8 & 91.2 \\
> \texttt{fastText}, $h=10$, bigram     && 92.5 & 96.8 & 98.6 & 95.7 & 63.9 & 72.3 & 60.2 & 94.6 \\
>
> From this result we can make the consistent conclusion with our paper.
> We have add multi-region size mode in our code and we will add more implement details in our paper, thank you for your suggestion!
> Is there a typo of the results you reported on Yahoo! Answers and Yelp Review? the numbers seems not similar with the results reported in this paper.

---

### Public Comment · ~Tobias_Engelhardt_Rasmussen1 · 2017-12-16
**Reproducibility review of "Bag og region embeddings via..."**

The authors behind the paper "Bag of region embeddings via local context unit for text classification" have come up with a new method to help classify text by looking at words' contextual effect on surrounding regions and how that changes with their relative positions. This is in contrast to just looking at the distribution of words and/or sequences of words. The authors were able to come up with some very good and state-of-the-art results. We have worked with the paper and the code made public by the team, in order to reproduce some of their results.

The report is very well structured, and it coherently develops the model contained within. Everything follows naturally with respect to these initial ideas. They show a lot of good examples of the power of their model, for example how some words contribute to the sentiment of the sentence, in different ways. The report does however have some grammatical errors, and some sections that did not read well.

We found that the results obtained by the authors were indeed reproducible, since we, using the same method and code, got very similar results. The authors made their code publicly available for all reviewers to see,  in the interests of reproducibility, which is definitely an asset for their credibility. Furthermore, the authors used publicly available data that was very easy to find. They were also able to compare their results to other more established models, which have all classified and obtained results on the same data. The authors tried two different approaches using the same model, and found that one was much more effective than the other, by comparing the results. One of their models got state-of-the-art results on a range of the datasets.

The Github code given by the authors had some good general instructions on how to run the pre-training and the actual training. Using Python 2.7.1 with Tensorflow and standard libraries such as Scipy and Numpy, we think that it is very accessible to reproduce their results, hardware considerations included.

The whole idea makes a lot of sense intuitively and mathematically, and we give the authors a lot of credit for this. They make it simple to understand. Even though the authors explained the feature selection model with good detail, and strong mathematical reasoning, we do not think the detail on the actual implementation is appropriate. We consider the lack of detail on the implementation a shortcoming of the paper, and the reproducibility of the results. The code given by the authors is very good, especially the hyper-parameter configuration file, which made it possible for us to reproduce the results. We did not find that we could run the code provided straight from Github, due to a very simple problem that was easily fixed. To make the code more easily understandable it would have been beneficial to add some more comments, since it is a non-trivial pre-training model, particularly given the lack of implementation detail in the paper. We do not think it would have been possible for us to reproduce the results without having been given the code.

Our results:

We tried to reproduce the results of the Word-context model which is the better one of the two models made by the authors. We tried only to reproduce the results on the two datasets, Yelp Full-Review and Yelp Polarity, and we obtained accuracies which were very comparable to the results published in the paper (within 0.5-0.8% of that published). We consider this result successful in our effort to reproduce the results. The key limitation of our work was the hardware, because the authors used very powerful hardware that was not available to us. With the hyperparameters given by the authors, and limited to one epoch with the hardware accessible to us, we found training to take around 12 hours. This forced us to slightly modify some of the hyperparameters, in order to run the code in a reasonable amount of time.

---

> ### Author Response · Authors · 2017-12-17
> **Reply**
>
> Thank you very much for the reproducing experiments. We have updated our code and discussed the common issues about the reproducibility.
> In summary, within 1% variance gap can be explained by the 90% training data in the published configure as default, while we use 100% in the paper results.
> The problem of slow convergence may be caused by the learning rate. In the example configure it was 1e-5 which declared 1e-4 in the paper.
> Please see our latest comments to get more detailed information.
> Thank you for your suggestions, we have refined the code with more guild-lines and comments.

---

### Public Comment · (anonymous) · 2017-12-16
**Reproducibility review**

This paper proposes two novel text classification models, Word-Context and Context-Word Region Embeddings, where every word is represented by a word embedding vector and a local context unit matrix. The local context unit is designed to capture the semantic and syntactic information of a word in a given context. The columns of the local context unit are used to interact with the embeddings of the words of the same region, creating projected word embeddings. Max pooling is applied to the projected word embeddings to create region embeddings which are context dependent. The Word-Context method is based on the interaction between the local context unit of a given word with the surrounding word embeddings, while the Context-Word is based on the interaction between the embedding of a given word with the surrounding local context units. A document is represented by a weighted summation of all its region embeddings, which is fed to an upper Fully-Connected layer for text classification. The proposed methods are designed to outperform some of the most commonly used methods in the literature such as bag of words, ngrams, bigram-FastText, D-LSTM among others.

The authors provide code to facilitate the reproduction of their results. Although mostly functional, it required some tweaking before it was ready to go. The code was used to reproduce the results found on the Yelp Full Review dataset, since the model's configuration parameters seem to have already been specified in the code. The results were found to be similar to those published by the authors (within 1\% variance). However, training the model can be computationally expensive if the required hardware is not available. Using a machine with 24 vCPUs and no powerful GPUs, the model required a training time of 1.2 hours per epoch. This presented serious constraints for tweaking the model's parameters.

Code for baseline implementations can be found in projects associated with the papers cited by the authors, but some of them have specific hardware requirements, and a modification of the input data format. The hyperparameters for implementing these baseline linear classifiers are in some cases left vague (such as the logistic regression of the n-gram TFIDF method). Thus, many baseline methods had to be inferred and implemented independently, and implementing the new methods required certain hardware resources to take advantage of the torch-driven parallelized implementation. It was thus challenging to reproduce the baseline results presented in this paper.

Finally, we introduce a new dataset on sentiment analysis of movie reviews from Kaggle to evaluate if this method can generalize well to other tasks. The dataset was particularly interesting since it had an imbalanced class distribution, and since it is associated to an online competition. The best published accuracy for this competition is 76.5\%. Although upon inspection this dataset had many noisy samples, word-context model resulted in a 54.02\% accuracy after twenty epochs of training, putting it slightly above the majority class prediction baseline. However, the test accuracy was constantly increasing until the last epoch, so if left longer the model might have achieved better classification results, or the convergence would have been faster if the parameters were tweaked. The main limitation here was again, computational complexity. We encourage the authors to participate in such competitions.

Strengths:
+ Intuitive concept, clear paper supported by figures and graphs
+ Open source code
Weaknesses:
- Code completeness and clarity
- Computational Complexity
Small Comments:
~ typos
~ Data statistics for the amazon polarity and full datasets in table 1 are interchanged

---

> ### Author Response · Authors · 2017-12-17
> **Reply**
>
> Thank you very much for the reproducing experiments. We have updated our code and discussed the common issues about the reproducibility.
>
> In summary, within 1% variance gap can be explained by the 90% training data in the published configure as default, while we use 100% in the paper results.
>
> The problem of slow convergence may be caused by the learning rate. In the example configure it was 1e-5 which declared 1e-4 in the paper.
>
> Please see our latest comments to get more detailed information.
>
> Thank you for your suggestions about participating in some competitions, we will consider it!

---

### Author Response · Authors · 2017-12-17
**Reproducibility official explanation**

Thank you very much for the effort on reproducing experiments and suggestions for this paper.

Although results on most datasets were reported reproducible, we have updated our code to reproduce more consistent experimental results straightly(including the exploratory experiments). Due to the time limit, the previous version of the shared code is not complete clear enough, we have update the code: 1)fixed a bug in the preprocess code which leads significant difference on DBPedia and AG News; 2)added exploratory experiments module, 3)published training configures for each dataset 4)added guild-lines and comments for the code. The latest code can be pulled from the same repository. We will also update this paper in a few days.

We reply issues about reproducibility here together:
1. Significant difference of reproducing results on DBPedia and AG News:
We find a bug in the public version of prepare.py that we treat the raw csv input files as two-columns for all datasets, while some of them are not.
This bug is caused by our negligence during migrate the code from internal version(which worked as expected) to public version. Unfortunately, we only verified the public version code on Yelp datasets which are two-columns files. This bug may lead the significant different reproducing results on DBPedia and AG News. We are very sorry for this bug in the shared code and now it has been fixed and verified reproducible.

2. 1% variance on most reproducing result:
Although similar results(variance within 1% on accuracy) have been reported on most datasets, the training data were defaultly set to 90% training data to tune the hyperparameters  in the example configure. Since we only tuned the hyperparameters on Yelp F and applied these hyperparameters on all datasets,  the results reported in this paper are trained by 100% training data, this can explain the variance on accuracy in reproducing results. And we have set 100% training data as default value in the new version of config.

3. Training time:
We have listed the training time and best epoch with different region sizes of each dataset in our paper(will be upload in a few days). In our experiments, it usually converges at 2 or 3  epoch with learning rate 1e-4 instead of more than 20 epochs, we are not sure whether it misled people that the initial learning rate in the example configure was 1e-5 which declared 1e-4 in the paper. Ignored the extra look up operation, the computational complexities of the proposed methods are basically the same magnitude with CNN. However, the shared code was not well optimized which may lead somehow slower in practice.

4. Hyperparameters:
We have tuned hyperparameters on Yelp. F. and applied them on all datasets, so there may be better hyperparameters for a given dataset. We chose the region size as 7 since we found that the performances with region size 7 and 9 were almost the same and 7 needed less model parameters, and similar with the embedding size.

5. Reproduction for baseline methods:
We have implemented some baseline models and achieved similar results with small fluctuations, considering the consistency of the comparison and lack of implementation details of some models, we reported the best results from previous works instead of reproducing all the baseline models.

---

### Author Response · Authors · 2018-01-03
**Submission Update 2017-01-03: Summary of Changes**

Thank you very much for reviewing our submission and making so many valuable comments. We also thank people for their attention to our work and their experiments in reproducing our results.

We have addressed all the issues pointed out by you and now the submission is of quite high quality.  Could you please review our submission again?  We hope that our submission will be accepted.

With the help from our colleagues, we have significantly improved the writing of the paper.  The title has been modified to better describe the main contribution of the work. The abstract and body have also been significantly revised accordingly.

We further plan to have a native speaker to conduct proof reading on our paper, if it is accepted.

Below is a summary of the major changes.

1. Title has been changed to “A new method of region embedding for text classification”.

2. Abstract and introduction have been deeply revised. The expression has been improved, and a more clear explanation about the local context unit has been added in the introduction.

3. In the related work section, discussion about "Attention is all you need" paper and citations including xx have been added.

4. In method, we have improved the notations. including the notations of projected embedding (e_{w_i}^j - > p_w{w_i}^j), region size(c -> 2 * c +1).  We also added equation numbers and refine the equations in 3.2 and 3.3.  More explanations and discussions about the intuition behind the approaches we produce the region embeddings have been added in 3.2.

5. We have added information about the datasets(average document lengths), implement details(multi-region-sizes mode), more cases about context units visualization, experimental results(training time and parameters numbers, best region sizes) in experiments section and appendix A. Figure 2 and Figure 3 are put next to each other to save space.

---

### Decision · Program_Chairs · 2018-01-29
**ICLR 2018 Conference Acceptance Decision**

**Decision:**

Accept (Poster)

**Comment:**

despite not amazing scores, this is a solid paper.
it created a lot of discussion and was found to be reproducible.
we should accept it to let the iclr community partake in the discussion and learn about this method of n-gram embeddings